Photogrammetric determination of movement speed of invasive Indo-Pacific lionfish in the Florida Keys

Kolonay Neal
Glaspie Cassandra N. cglaspie1@lsu.edu
Department of Oceanography and Coastal Sciences, Louisiana State University and Agricultural and Mechanical College , Baton Rouge , LA , United States of America
Zucchetta Matteo
Electronic publication date: 2025 Jan 20
Publication date: 2025
Volume: 13
Electronic Location ID: e18474
Received 2023 Oct 9; Accepted 2024 Oct 15
Copyright: ©2025 Kolonay and Glaspie al.
Copyright year: 2025
Copyright holder: Kolonay and Glaspie
License: This is an open access article distributed under the terms of the Creative Commons Attribution License, which permits unrestricted use, distribution, reproduction and adaptation in any medium and for any purpose provided that it is properly attributed. For attribution, the original author(s), title, publication source (PeerJ) and either DOI or URL of the article must be cited.
License URL: https://creativecommons.org/licenses/by/4.0/

Keywords: Pterois miles, Pterois volitans, Swimming speed, Underwater photography, Stereo-camera, Behavior, Fish, Foraging, Reef

Funding: the Louisiana Board of Regents through the Board of Regents Support Fund LEQSF(2019-22)-RD-A-06 This work was supported by the Louisiana Board of Regents through the Board of Regents Support Fund, Contract No. LEQSF(2019-22)-RD-A-06. The funders had no role in study design, data collection and analysis, decision to publish, or preparation of the manuscript.

==============================
As a key determinant of how efficiently lionfish (Pterois sp.) locate and capture prey, swimming speed plays a crucial role in shaping the predator-prey interactions and broader ecological dynamics within the invaded ecosystems. Swimming speed on a small temporal and spatial scale is difficult to measure because of the need for precise measurements of both distance and duration of the behavior. Using photogrammetry by way of stereo-camera setups is ideal for analyzing the minutiae of lionfish behaviors because it can include the benefits of remote video traps coupled with precise measurements of movements in three-dimensional space and time. The primary objective of this study was to identify and characterize lionfish behavior associated with different movement speeds, and then to quantify small-scale swimming speeds of lionfish associated with those behaviors. Swimming speeds were classified under three different observed behaviors: relaxed swimming, traverse swimming, and striking at prey. The differences between these behaviors were primarily distinguished based on body and fin positioning, as well as the apparent intent of the motion if any was evident. The mean lionfish swimming speed from stereoscopic camera footage was 44.75 mm s−1 for relaxed swimming, 138.99 mm s−1 for traverse swimming, and 625.44 mm s−1 for striking at prey. Swimming speed can be used to quantify how much habitat area a lionfish may cover in a day, and therefore the amount of prey that may be encountered by a predator. Lionfish feeding success under different environmental conditions could be an important factor in understanding their survival and growth in areas where they are found.

Introduction

Lionfishes, both Pterois volitans and P. miles (hereafter referred to as lionfish) were most likely introduced by aquarium keepers off the coast of southeast Florida and their established range now reaches from New York, through the Gulf of Mexico, the Caribbean, and as far south as Brazil (Schofield, 2010; Ferreira et al., 2015). More recently, lionfish have invaded and are now established along the southern coast of the Mediterranean Sea (Kletou, Hall-Spencer & Kleitou, 2016). High densities of lionfish can cause a loss of over 97% of prey-sized native fishes in just 7 weeks on reefs where they are found due to high feeding rates (Benkwitt, 2015) and can reduce recruitment of native species by an average of 79% due to predation on juvenile or larval fishes (Albins & Hixon, 2008). This can cause a loss of biodiversity as well as decrease the abundance of large predators such as snapper and groupers, both due to predation on juveniles as well as competition for the prey that these larger mesopredators target (Morris Jr & Whitfield, 2009). Lionfish have been successful in their invaded range for several reasons. One major reason is that although there are many potential predators of lionfish in the invaded range (Ulman et al., 2021), the presence of potential piscivorous predators does not significantly change lionfish density in that range (Cure, McIlwain & Hixon, 2014), indicating that there is not enough predation pressure to keep the lionfish population under control. Two other traits of lionfish that contribute to their success as invaders are their generalist and indiscriminate diet and their successful hunting (Peake et al., 2018). These fish can consume large amounts of food in a short time, with their stomachs extending up to 30 times the original size (Morris Jr & Whitfield, 2009) and consuming an average of 14.6 g day−1 as adults of 300–500 g body weight (Fishelson, 1997). As a result, lionfish in their invaded range grow faster than their native counterparts (Pusack et al., 2016). Each adult female lionfish is also capable of spawning every 2.5 days, year-round, with an annual output of over two million eggs (Fogg, Brown-Peterson & Peterson, 2017); all of these factors make this species a devastating invader.

Lionfish tend to establish a home range around a specific reef structure, the estimated size of which varies widely between different locations and methods. Jud & Layman (2012) used a mark-recapture study and concluded that the majority of individuals stay within 10 m of their capture site in a Florida estuary. Green et al. (2021) used acoustic tagging to estimate the lionfish home range in the U.S. Virgin Islands to be much larger, with a radius of 180 m. Despite their tendency to stay within a certain area, lionfish can move large distances (2–10 km) to colonize other reefs (Dahl & Patterson, 2020; Green et al., 2021). Given the ability of lionfish to consume large quantities of reef fish, it is necessary to understand the movement of lionfish to be able to model their impacts to invaded reef communities.

Foraging models provide a framework for assessing the influence of predation on ecosystems. By considering variables such as prey abundance, predator search behavior, and prey vulnerability, foraging models allow researchers to predict the potential impact of fish predators on prey populations (Brandt, Mason & Patrick, 1992). In a foraging model, swimming speed is a key parameter in the calculation of search volume, a critical concept in understanding the foraging behavior of fish and their potential impact on prey populations (Boisclair & Sirois, 1993; Brownscombe et al., 2014). A higher swimming speed results in a larger search volume, meaning the fish encounters a greater expanse of the surrounding habitat within a given time frame. This, in turn, translates to an increased likelihood of encountering potential prey items. In the context of lionfish predation, a faster swimming speed allows them to explore more territory, effectively expanding their hunting grounds and encountering a higher number of potential prey organisms. As a key determinant of how efficiently lionfish locate and capture prey, swimming speed plays a crucial role in shaping the predator–prey interactions and broader ecological dynamics within invaded ecosystems.

Lionfish movement speed has been quantified in a few different ways, each with different strengths and weaknesses. Large-scale movement has been quantified using acoustic tagging and telemetry (Bacheler et al., 2015). While telemetry is important in understanding average swimming speed over larger temporal and spatial scales, it is not able to resolve movements at the scale of an individual reef. Diver surveys are useful for understanding the behavior of individual lionfish, including movements on the scale of meters, but they have several limitations: they potentially disturb the behavior of the organism being analyzed, are very limited in their time and depth, and lack accuracy and precision (Rutecki, Schneeberger & Jude, 1983). Lionfish pursue prey on a spatial scale of cm to m; to model lionfish foraging behavior, a strategy is needed to characterize lionfish movement at that scale.

Swimming speed on a small temporal and spatial scale is difficult to measure because of the need for precise measurements of both distance and duration of the behavior. These types of movements are typically measured in a laboratory setting (Peterson & McHenry, 2022), but fish often do not exhibit natural behaviors in captivity. Remote camera systems have little to no disturbance on fish behavior and can be deployed in-situ for long durations and without depth limitations (Rutecki, Schneeberger & Jude, 1983; Harvey, Fletcher & Shortis, 2002; Cappo et al., 2003; Harvey et al., 2004; Mills, Verdouw & Frusher, 2005). In addition, swimming speed often differs with behavior, such as foraging, mating, or fighting (Price, 1989; Brownscombe et al., 2014; Marras et al., 2015); remote camera deployments allow researchers to assess visual cues associated with fish behavior and characterize movement speeds associated with different behaviors.

Using photogrammetry by way of stereo-camera setups is ideal for analyzing the minutiae of lionfish behaviors because it can include the benefits of remote video traps coupled with precise measurements of movements in three-dimensional space and time. Photogrammetry, the science of extracting quantitative information from photographs or digital media, is not a novel concept, and various forms of it have been in use for many years (Doyle, 1964). In a stereo-camera or stereoscopic system two cameras are attached to a bar with a known separation distance and angle of convergence (Letessier et al., 2013), which allows measurements in three dimensions with respect to the cameras with each pair of photographs taken. A stereo-video system allows measurements to be taken with respect to time as well as space. A fish can be within a large range of distance of the camera system, and at nearly any angle or orientation, and distances can still be measured on the scale of cm. This study used stereo-video camera systems to examine the behavior of lionfish cohabiting shallow patch reefs in the Florida Keys, USA. The primary objective of this study was to identify and characterize lionfish body and fin positioning associated with different movement behaviors, and then to quantify small-scale swimming speeds of lionfish associated with those behaviors.

Materials & Methods

Portions of this text were previously published as part of a thesis (Kolonay, 2022).

Camera system

The camera system used in this project was custom built by SeaGIS (http://www.seagis.com.au/). The camera mount was an aluminum bar 75 cm in length, and 10 cm wide, with angled housings attached on each end that pointed the cameras toward each other so that their field of view overlapped (Fig. 1). When deployed in the water, the camera system housing was bolted to a frame made from steel, with a width of 40.5 cm, a length of 88 cm, and a height of 53 cm. The purpose of the frame was to weigh down the camera system to keep it on the bottom, and to keep the cameras stable and off the sand.

Figure 1 SeaGIS stereo camera deployment.

Cameras were angled toward a patch reef to allow us to track movement in three-dimensional space.

The cameras used were Canon HFG10 models, 1080p resolution, with settings mode set to M and FXP recording. A 60i frame rate was used leading to a working frame rate of 29.97 frames per second. Canon brand lithium-ion batteries were used to extend recording times. For calibration purposes six total video cameras were enclosed in paired underwater housings and brought to a pool with the SeaGis 3D calibration cube, where initial footage was taken to be processed in the SeaGis CAL™ software. In this software, bundle adjustments were made for calibration, and camera files were obtained for each individual camera system. Final video footage from the field was combined with the calibrated camera files, and the SeaGis CAL™ software EventMeasure™ was used to synchronize the two video files and allow measurement in 3D (Letessier et al., 2013).

Site selection and videography

Unbaited stereoscopic camera systems were used to measure lionfish movement speed on shallow patch reefs in the Florida Keys, USA. Camera systems were deployed near a patch reef in the Florida Keys off Summerland Key and near Looe Key Sanctuary Preservation Area located within the Florida Keys National Marine Sanctuary. The patch chosen for filming had approximately 2–6 adult lionfish on the patch reef consistently throughout the study period. The patch reef used was at 24.537°N, −81.435°W, at 15–16 m depth, with mean water temperature 28.5 °C and no discernable current. Visibility generally ranged between 20 and 30 m throughout the study period, but never less than 15 m, even after significant weather events that produced high seas (1 m) and rainfall that results in runoff that generally reduce visibility. The patch reef was roughly 15 m across, 5 m wide, and 2 m in height, with high rugosity and structural complexity. There were both corals and sponges present on the patch reef, but they were sparse; most of the patch reef substrate was rocky outcrop and algae. The patch reef was surrounded with sand and patchy rubble, and the nearest patch reefs were roughly 30 m away in any direction. Lionfish prey such as small crustaceans, damselfishes, and wrasse were abundant. This patch reef was identified as an ideal location for observations due to the reoccurring presence of lionfish that had high site fidelity, thus maximizing potential behavior observations via camera deployment.

Patch reefs were filmed before dusk for a period of roughly 3 to 4 h (limited by the battery capacity of the cameras) and recovered the next morning. Three camera systems with two cameras each were used to collect data each deployment, and several deployments were made in different areas to capture footage of different fish. Five days of footage were collected from June 12th to June 17th, 2021.

Behavior characterization

Swimming speeds were classified under three different observed behaviors: relaxed swimming, traverse swimming, and striking at prey. The differences between these behaviors were primarily distinguished based on body and fin positioning, as well as the apparent intent of the motion if any was evident. Durations of measurement varied but were typically less than one minute long. Durations were judged by when fin positioning movement changed fully to the described behaviors to when body momentum ceased, and fin positioning changed again. Fin positioning included primarily dorsal spines and pectoral fins, which are further broken down into dorsal, medial, and ventral segments.

Relaxed swimming was characterized by a very slack, low effort, forward swim, with a neutral positioning of all fins (Fig. 2). Dorsal spines were near fully or fully erect, and pectoral fins lay primarily to the side of the body, neither flared nor raised. The dorsal section of the pectoral fin was typically near the midpoint of the fish both horizontally and vertically. The medial portion of the pectoral fin was typically parallel or close to parallel to the body of the fish. This was the position that had the widest variation, though the low frequency at which the caudal fin moved and the lack of any noticeable object or prey the body was pointing towards also helped to classify a swim as relaxed.

Figure 2 Examples of relaxed swimming in lionfish.

Main image: two lionfish exhibiting relaxed swimming with all fins in a neutral position with no dorsal fin tucking and no flaring of pectoral fins. Inset: A frontal view of a lionfish exhibiting relaxed swimming.

Traverse swimming was distinct from relaxed swimming in that the dorsal spines as well as all three parts of the pectoral fins were completely folded back against the body, and the frequency of caudal fin movement increased rapidly, propelling the fish forward (Fig. 3). This type of swimming appeared to be intended to traverse small distances very quickly though the reason was not always clear, and not always captured in the footage. The body position for traverse swimming typically included a sudden pivot away from where the lionfish was currently starting, and the end point if visible was either shelter, an object, or another fish (though not always a prey-sized organism). While rare, this behavior seemed to be used to escape much larger organisms, potentially those seen as a threat to the lionfish, such as approaching human divers.

Figure 3 An example of traverse swimming in lionfish.

In traverse swimming, both dorsal and pectoral fins are folded into the body with rapid beats of the caudal fin.

Striking at prey was always preceded by a hunting behavior which included two types of fin and body positioning. The body position of a lionfish striking at prey was almost always angled directly at a prey organism, leading often to a perpendicular body position with reference to the substrate, rock, or coral head. The dorsal spines were erect for lionfish striking at prey. The fin position altered between (1) a full flare of the pectoral fin vertically perpendicular to the body, where most or all of the three portions of the pectoral fin were extended in a vertical semi- circle on either side of the body (Fig. 4); and (2) a horizontal flare of the dorsal portion of the pectoral, with the medial portion tucked against the body, and the ventral either pointing down or also tucked against the body (Fig. 4). See Kolonay & Glaspie (2024) for further characterization of foraging behavior and videos of fish exhibiting these behaviors.

Figure 4 Three lionfish exhibiting foraging behavior preceding a strike on prey.

Preceding a strike, lionfish flare dorsal and pectoral fins both horizontally and vertically. A hunting lionfish will follow with a quick forward strike, angling the body towards a prey fish, generally with substrate (reef) behind the prey.

The prey strike itself consisted of a rapid lunge towards the prey with a swift beating of the caudal fin, while dorsal spines and pectoral fins were folded back against the body, similar to the traverse swim. Key differences between this behavior and the traversal swim were the hunting behaviors described above preceding the strike itself, the presence of an apparent prey item or items, and a much shorter distance and duration of the swim (often only a few frames). These differences can also be used to differentiate prey strikes from other similar short and quick movements, such as after yawning and during aggression.

Swimming speed

Swimming speeds were calculated using measurements from previously calibrated video footage using the program EventMeasure™. Some of the footage was internally edited in EventMeasure™ to change the brightness and contrast, making measurements easier to obtain. These were temporary edits, and did not affect the measurements themselves, as timing and scale of the footage remained unchanged. The benefit of photogrammetric measurements is that movement in any direction can be measured. However, when attempting to calculate a speed at which direct swimming movements occur, large amounts of erratic lateral movement, or winding paths will not give an accurate idea of the speed of the movement in one direction, regardless of which direction that may be. Thus, to reduce the error from winding or non-direct paths, an attempt was made to primarily measure straight line movements.

When an adult lionfish was observed to swim unobstructed across the camera field of view, a segment in which the lionfish was traveling in a straight line was used to calculate swimming speed. The location of the lionfish at the beginning and end of the swim was determined as the midpoint of the lionfish, as determined by a direct line from the center of the eye to the middle of the last stripe of each fish. This measurement was used because the caudal fin of lionfish is both translucent, making it hard to capture in some video, as well as easily damaged, meaning that not all lionfish have equally proportioned caudal fins relative to their body size. Measuring from the eye to the middle of the final stripe allowed easier and more consistent measurements than attempting total length. The midpoint of this measurement was used to account for small variations in body angle and movement of extremities. Measuring solely the eye, tip of mouth, or other landmarks on the fish would give less accurate measurements of whole-body movement.

For prey strikes, although the mouth does move independently of the body as the jaw protrudes, the movement of the mouth was not measured independently due to limitations of the camera system, and only the whole-body movement was measured. Rarely, the eye to final stripe measurement was not possible, in which case a measurement from the eye to the center of an easily visible stripe was taken, and the same measurement was used in each frame.

Measurements taken in the software were exported to excel, and the 3D distance formula combined with the division of time was used to calculate the swimming speed: (1) Swimming speed=x2−x12+y2−y12+z2−z12t2−t1

where x1, y1, and z1 were the initial midpoint position of the lionfish in millimeters, and x2, y2, and z2 were the final midpoint positions of the lionfish in millimeters, t1 was the time when the fish was present at the initial point in seconds, and t2 was the time when the fish was at the final point in seconds. The top portion of this equation yielded the distance in millimeters traveled by the lionfish, and the bottom half yielded the amount of time in seconds the lionfish took to travel the distance, giving in total the speed at which it was swimming in mm s−1.

Each event used to calculate swimming speed was not a unique lionfish; given the number of individual fish on the reef, it is likely that the data collected were repeated measurements of six individual adult lionfish (>200 mm standard length). However, it was not possible to identify individuals from most video segments, so each datum is treated as independent for the purposes of this analysis.

Analysis

A Shapiro test and a Levene test were run to assess the ANOVA assumptions of normality and homogeneity of variance. A square root transformation was applied to swimming speeds to meet the assumption of homogeneity of variance. An ANOVA was performed with square root transformed swimming speed as the response and type of swimming behavior (relaxed, transverse, and strike) as the predictor variable. A Tukey HSD post hoc test was used for pairwise comparisons.

Ethics statement

The Louisiana State University Institutional Animal Care and Use Committee provided full approval for this research (IACUC Protocol #19-050). Field experiments in the Florida Keys National Marine Sanctuary were approved by the Florida Fish and Wildlife Conservation Commission under a Special Activity License (SAL-20-2250-SR), and The National Oceanic and Atmospheric Administration Office of National Marine Sanctuaries under permit number FKNMS-2020-087.

Results

Over 1,000 min of video was collected over five deployments of the camera system. A total of 42 instances of lionfish swimming behavior were identified and analyzed for swimming speed.

The results of the stereoscopic camera footage show how lionfish move during three different behaviors, relaxed swimming, traverse swimming, and striking at prey. More relaxed swims (n = 15) were captured than traverse swims (n = 12) and strikes (n = 5), as they were by far the most common of the swimming behaviors. The mean swimming speeds and standard deviations for each behavior were as follows: relaxed swimming 44.75 ± 15.46 mm s−1, traverse swimming 138.99 ± 48.66 mm s−1, and strike 625.44 ± 66.94 mm s−1.

The ANOVA provided strong evidence for a significant difference among swimming speeds associated with each type of swimming behavior identified (F2,39 = 301.5, p < 0.0001; Fig. 5). The post hoc analysis identified a difference in swimming speeds for all pairwise comparisons (relaxed versus traverse, p < 0.0001; relaxed versus strike, p < 0.0001; traverse versus strike, p < 0.0001).

Figure 5 Violin boxplot of lionfish swimming speed.

The width of the “violins” denotes the distribution of observations. Small overlap occurs between untransformed swimming speeds for relaxed and traverse, but neither overlap with strike speed.

Discussion

This study represents the first time that small scale lionfish movement speed during different behaviors has been measured with the use of photogrammetry. The lionfish in this study demonstrated three distinct swimming behaviors on patch reefs in the Florida Keys. Relaxed swimming, traverse swimming, and strikes all had significantly different speeds when compared to each other. There was minor overlap between statistical outliers of relaxed and traverse swimming, but no overlap between statistical outliers of strikes and those of any other behavior.

Previous literature on lionfish movement speed is primarily focused on large scale movements between areas of reefs on a time scale longer than this study, often by acoustic telemetry or tagging (Bacheler et al., 2015; Tamburello & Côté, 2015; McCallister et al., 2018). Two recent papers by Green et al. (2021) and Peterson & McHenry (2022) provide literature values for small scale lionfish movement speed; the measurements of relaxed lionfish swim speed (average = 44.75 mm s−1) in this study are close to the ranges presented in these previous studies, being similar to the Peterson & McHenry (2022) average of 64 mm s−1 for lionfish in the laboratory and similar to the average of tagged fish in Green et al. (2021) of 51 mm s−1. Green et al. (2021) also noted the several individuals that traveled distances >1 km had an average swim speed of 260 mm s−1, which is closer to the maximum of the traverse swimming speed presented in this study at 209 mm s−1. Both studies noted movement over longer periods (10 to 30 min) than this study, which measured individual movement segments over periods of <1 min. Given the length of time in previous studies, it is likely that these were not straight line measurements as in this study, and they may have included multiple behaviors, such as relaxed swimming and hunting, as well as assisted movement due to prevailing currents, all within the measurement period.

Although it may seem ideal to observe natural fish behavior in-situ using deployed camera systems, this method has some disadvantages. The resolution of the video did not allow us to distinguish among individual fish, so it is likely that pseudoreplication occurred in our data in which the same fish was observed multiple times. These data are treated as replicates in our analysis, and this choice may lead to false precision and increased likelihood of a Type I error. In addition, the framerate of 29.97 fps, while sufficient for most swimming speeds, may have been too low to capture the high acceleration and rapid movements that occur during a strike on prey. The speed associated with strike behavior in this study should be treated with caution, as the estimate may lack precision.

The data obtained in this study are useful in several areas, but primarily as a parameter for realistic foraging models to support management of invasive lionfish. Different behaviors are associated with different swimming speeds, and the swimming speed can be used in bioenergetics models to determine how energy is expended over time as well as how an individual fish approaches gaining energy (Price, 1989; Boisclair & Sirois, 1993; Brownscombe et al., 2014; Marras et al., 2015). Swimming speed data paired with large scale acoustic telemetry and tagging (Bacheler et al., 2015; Tamburello & Côté, 2015; McCallister et al., 2018) and input into modeling efforts such as spatially-explicit bioenergetics models (Cerino et al., 2013) could be useful for planning how lionfish interact with their environment, and how changing conditions might impact the lionfish population in a given area.

Conclusions

In-situ swimming speed of different behaviors has not been well studied in lionfish, and a better understanding of lionfish movement could provide valuable insight to their foraging behavior. This study measured in situ swimming speed of lionfish on patch reefs in the Florida Keys. Posture cues were identified to distinguish three different swimming behaviors: relaxed swimming, traverse swimming, and strikes. Swimming speeds for lionfish exhibiting relaxed and transverse swimming from in-situ videos were similar to swimming speed determined in the laboratory and from acoustic tagging studies of large-scale lionfish movement. Swimming speed can be used to quantify how much habitat area a lionfish may cover in a day, and therefore the amount of prey that may be encountered, allowing for models of lionfish foraging to be built and applied to the management of this invasive species.

Supplemental Information

Supplemental Information 1 Lionfish swimming speed associated with different behaviors

Column 1 ”Speed” is measured lionfish speed in mm s-1. Column 2 ”Type” is the type of swimming behavior exhibited by the lionfish when the speed was measured, either relaxed, traverse, or strike.

We thank James Seager for his great assistance and guidance on the setup and use of stereo camera systems, and the many students and volunteers from the Marine Community Ecology lab at LSU, especially Emily Shallow and Calvin Glaspie for their help during field work. Special thanks to Lana Neff, for her continual support.

Additional Information and Declarations

Competing Interests

Author Contributions

Animal Ethics

Field Study Permissions

Data Availability

The authors declare there are no competing interests.

Neal Kolonay conceived and designed the experiments, performed the experiments, analyzed the data, prepared figures and/or tables, authored or reviewed drafts of the article, and approved the final draft.

Cassandra N. Glaspie conceived and designed the experiments, performed the experiments, prepared figures and/or tables, authored or reviewed drafts of the article, and approved the final draft.

The following information was supplied relating to ethical approvals (i.e., approving body and any reference numbers):

The Louisiana State University Institutional Animal Care and Use Committee provided full approval for this research. (IACUC Protocol #19-050)

The following information was supplied relating to field study approvals (i.e., approving body and any reference numbers):

Field experiments in the Florida Keys National Marine Sanctuary were approved by the Florida Fish and Wildlife Conservation Commission under a Special Activity License (SAL-20-2250-SR), and The National Oceanic and Atmospheric Administration Office of National Marine Sanctuaries under permit number (FKNMS-2020-087).

The following information was supplied regarding data availability:

The raw measurements are available in the Supplemental File.

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
