# Peer review of "Photogrammetric determination of movement speed of invasive Indo-Pacific lionfish in the Florida Keys"

_PeerJ, doi:10.7717/peerj.18474_

## Round 0.1 · original submission · Minor Revisions

As you can see, the manuscript was generally well-received by the reviewers. Please consider also the comments of Reviewer 1, who suggested rejecting the manuscript, as they contain some valuable indications for improving the paper.

·

Basic reporting

The authors present a method for studying small scale movements of lionfish, but I have major concerns regarding their analyses and lack of detail in the current manuscript (see sections 2 and 3 for more detail). It is likely that the authors have a good video dataset, and the current topic would certainly be of interest to those in ecology and biology, but I feel that the manuscript in its current state is not fit for publication.

The introduction has weight and contains many relevant citations, but there doesn't seem to be a clear hypothesis, nor clear defined aims. I am concerned that the current analysis is not rigorous enough to capture meaningful data and is not detailed enough to assess issues of sample size, pseudoreplication, ect. The results and discussion sections need additional revision. For me, five sentences of a result section is not sufficient. I was surprised that the only data reported here is a single average speed value and observed behaviors, I would love to see more data from 3D insitu recordings. The paper is readable, but would greatly improve with some careful editing.

Experimental design

I have observed the changing fin/body configurations described, and am happy to see it being described for publication, but I have concerns about the way speed was calculated and the way in which the data was binned. The methods suggest that observed swimming behavior is a metric used (amongst others) to bin behaviors into the three categories, so it is unsurprising that the statistics show a significant difference, as the act of sorting these behaviors may inherently rely on speed. Additionally, from my understanding the speed reported here is actually an average speed for each behavioral 'instance' measured. I don't understand why the authors would not report the instantaneous speed as a function of time, then calculate the average of that. The fact that there are such large differences in speed means there were large changes in displacement, so variable speeds over time may not be accounted for in the measurements reported here. Lastly, while 29.97 fps is sufficient for routine swimming, any strike that was recorded at this frame rate would likely only show up in a tiny number of frames due to the high acceleration and rapid movements that occur during suction feeding. I'm not sure speed could confidently be calculated for this type of behavior without filming at a higher frame rate.

I have additional concerns about the number of samples taken. Was each data point a single independent individual, or were there multiple data points from the same individual? Where there multiple behavioral states for the same individuals?

This information is needed to fully asses the quality of the study.

Validity of the findings

The authors have included both a table and a plot of the same data. I would rather see an improved plot and the raw data listed in the results as mean plus or minus standard deviation (or standard error), as seems to be the standard for kinematics papers. I would highly suggest the authors review a few papers from different journals that focus on descriptive animal kinematics.

Figure suggestions.
Table 1: The table seems unnecessary as summary values can be included in the results, with references to figure 5.

Figures 2-3: The authors may benefit from including a schematic/drawing of the fin positions, and using the photos as examples. This could likely be done in one figure with many panels. I would recommend cropping the images, as the reef background is quite nosy and the lionfish are relatively small in frame. The images also seem very grainy for 1080, I am unsure if this is an artifact of the pdf form, but it would be great if higher res images were available.

Figure 5: The inclusion of both boxplots and violin plots is confusing and unnecessary. I would pick one and stick with better descriptions of what the components are. In my pdf version there is no axis lines, and the text of the axis ticks is huge relative to the axis titles. I assume this is the untransformed data, but to me the distribution of the related data looks normal, while the traverse and strike data looks very skewed/non-normal.

Additional comments

While I have major concerns with the analysis and the manuscript in it's current state, I want to make clear that I look forward to seeing the data from this study in a future paper. I have no doubts that the authors have a good dataset that would be of interest to many.

·

Basic reporting

I like very much this manuscript. Its subject has not been explored in depth recently.
Also, it is short, remarking the findings and forgetting about long wording.

Experimental design

It is clearly explained and can be replicated.

Validity of the findings

In my opinion their findings are intersting and very valuable.

Additional comments

The manuscript should be published.

Reviewer 3 ·

Basic reporting

See attached PDF

Experimental design

See attached PDF

Validity of the findings

See attached PDF

Additional comments

See attached PDF

Annotated reviews are not available for download in order to protect the identity of reviewers who chose to remain anonymous.

·

Basic reporting

This is a well written and important study with regards to lionfish and how important swimming speed is in determining how effective their predator prey interactions are. I commend you for developing an up to date base line study especially where there is a gap in our knowledge.

Experimental design

The design is well thought out and definitely goes a long way to fill in some gaps in knowledge, creating a starting point for future research.

Validity of the findings

These findings are important. I would like some clarification however on a few points. When you studied the behaviour did you look at 42 individual lionfish or was it different behaviours from the same lionfish? You need to clarify if the results came from different individuals or if the lionfish were reused. Did their behaviours change? If so did you take the first behaviour or include any others? How did you account for the possibility the fast current may have been masking transverse behaviour?

Additional comments

I think once you clarify the comments I made it will ensure anyone reading this in the future will not be left wondering the same things.

·

Basic reporting

-The article is well written and clear. It follows professional standards.
-The article has updated literature relevant to the topic of study. The introduction and justification of the work are correctly developed.
-The structure of the article follows the format of the journal.
-Although the figures and the table are relevant to the study, I consider it necessary to improve the resolution of them, especially in Figure 4. It is difficult to identify the angle of the pectoral fins of the individuals.
-I recommend adding short videos (such as supplemental files) with swimming movement types.

Experimental design

-It represents an original, relevant and scientifically rigorous article. The research question is clear. The results help to fill a knowledge gap.
-Research follows a high technical and ethical standard.
-I believe that the methodology can be improved. It is not known how many individuals were counted and whether they correspond to adults or juveniles, it is necessary to add the range of the standard length of the individuals.
This is important to clarify as some studies support an ontogenetic change in the diet of lionfish, where small sizes or juveniles (<200 mm) tend to consume more invertebrates (mostly crustaceans) and as the individuals grow, they become primarily piscivorous (Morris & Akins, 2009; Gómez-Pardo, 2014; Romero, 2017; Acero P. et al., 2019).
Possibly movement behavior and swimming speeds also change along the lionfish ontogeny.

Validity of the findings

-Data is strong and statistically supported
-The conclusions are clear and answer the research question

---

## Round 0.2 · accepted · Accept

Although one of the reviewers remains unconvinced about the manuscript's suitability for publication, based on the opinions of the other reviewers (on this and the previous round of review), your responses, and the improvements made to the manuscript, I believe it is now ready for publication.

·

Basic reporting

I appreciate the work of the authors to respond and update their manuscript. I do not doubt the authors have a good video dataset, and I think the descriptions of swimming behavior is of interest to the readership of the journal. However, my major concern is still that the data presented is insufficient to address the aims and expectations presented in the manuscript in a quantitative manor.

The authors have clarified that measured swimming speed itself was not used to characterize the behavior of prey. However, it is difficult to untangle speed and swimming behavior, so I still believe that the classification of behavior itself plays a part in the statistical significance of swimming speeds. The authors themselves state that the frequency of the caudal fin beating was a part of the behavioral characterization.

As mentioned previously, I also have issues with the measurement of strike speed, that I do not believe can be sufficiently addressed here. If the authors chose to exclude the actual suction feeding event, and instead looked at speed during the pursuit phase, where the fins are erect, I suspect they would get much more similar numbers to the other speeds. I believe the high speeds reported for a strike are skewed by the very short period of acceleration that occurs during the suction feeding event, and are not representative of the swimming behavior that occurs during the ‘hunting’ period. The authors themselves have admitted to this, but I do not agree with their decision to leave the data in.

The authors do mention that this manuscript is designated as a ‘note’, this was not something I was aware of. As a reviewer the manuscript is designated as a research article. I do feel the data presented here is more suited to a ‘bulletin’ or ‘note’ type manuscript. In which case the manuscript could be presented in a more succinct and qualitative manner, reducing speculation of statistical errors that come from potential psuedoreplication and low spatial/temporal resolution (in the case of the strike category).

Experimental design

no comment

Validity of the findings

no comment

Additional comments

no comment

·

Basic reporting

no comment

Experimental design

no comment

Validity of the findings

no comment

Additional comments

no comment

·

Basic reporting

The manuscript is clear, concise and well written. Added relevant references.

Experimental design

The methodology was clarified significantly.

Validity of the findings

Although the authors mention some methodological limitations (pseudoreplication), the article is relevant, innovative and well structured.

Additional comments

No comments